# JNK and Yorkie drive tumor malignancy by inducing L-amino acid transporter 1 in *Drosophila*

Bojie Cong[1], Mai Nakamura[1], Yukari Sando[1], Takefumi Kondo[2,3], Shizue Ohsawa[4], Tatsushi Igaki[1] *

**1** Laboratory of Genetics, Graduate School of Biostudies, Kyoto University, Yoshida-Konoe-cho, Sakyo-ku, Kyoto, Japan, **2** Graduate School of Biostudies, Kyoto University, Yoshida-Konoe-cho, Sakyo-ku, Kyoto, Japan, **3** The Keihanshin Consortium for Fostering the Next Generation of Global Leaders in Research (K-CONNEX), Sakyo-ku, Kyoto, Japan, **4** Group of Genetics, Division of Biological Science, Graduate School of Science, Nagoya University, Furocho, Nagoya Chikusa-ku, Aichi, Japan

* igaki.tatsushi.4s@kyoto-u.ac.jp

**Data Availability Statement:** All relevant data are within the manuscript and its Supporting Information files.

## Abstract

Identifying a common oncogenesis pathway among tumors with different oncogenic mutations is critical for developing anti-cancer strategies. Here, we performed transcriptome analyses on two different models of *Drosophila* malignant tumors caused by Ras activation with cell polarity defects (Ras$^{V12}$/*scrib*$^{-/-}$) or by microRNA bantam overexpression with endocytic defects (bantam/*rab5*$^{-/-}$), followed by an RNAi screen for genes commonly essential for tumor growth and malignancy. We identified that Juvenile hormone Inducible-21 (JhI-21), a *Drosophila* homolog of the L-amino acid transporter 1 (LAT1), is upregulated in these malignant tumors with different oncogenic mutations and knocking down of JhI-21 strongly blocked their growth and invasion. JhI-21 expression was induced by simultaneous activation of c-Jun N-terminal kinase (JNK) and Yorkie (Yki) in these tumors and thereby contributed to tumor growth and progression by activating the mTOR-S6 pathway. Pharmacological inhibition of LAT1 activity in *Drosophila* larvae significantly suppressed growth of Ras$^{V12}$/*scrib*$^{-/-}$ tumors. Intriguingly, LAT1 inhibitory drugs did not suppress growth of bantam/*rab5*$^{-/-}$ tumors and overexpression of bantam rendered Ras$^{V12}$/*scrib*$^{-/-}$ tumors unresponsive to LAT1 inhibitors. Further analyses with RNA sequencing of bantam-expressing clones followed by an RNAi screen suggested that bantam induces drug resistance against LAT1 inhibitors via downregulation of the TMEM135-like gene *CG31157*. Our observations unveil an evolutionarily conserved role of LAT1 induction in driving *Drosophila* tumor malignancy and provide a powerful genetic model for studying cancer progression and drug resistance.

## Author summary

Cancers develop through the accumulation of multiple oncogenic mutations. Such mutations are different among tumors of different origins and thus identification of a common

**Funding:** This work was supported by Grant-in-Aid for Scientific Research (A) (Grant No. 20H00515) to T.I, Japan Agency for Medical Research and Development (Project for Elucidating and Controlling Mechanisms of Aging and Longevity; Grant No. 21gm5010001) to T.I, Grant-in-Aid for Scientific Research on Innovative Areas (Grant No. 20H05320) to T.I, Grant-in-Aid for Transformative Research Areas A (Grant No. 20H05945 and 21H05284) to S.O and T.I, JST Moonshot R&D (Grant No. JPMJMS2022) to S.O, the Naito Foundation to S.O and T.I, the Takeda Science Foundation to S.O and T.I, Toray Science Foundation to S.O, and MSD Life Science Foundation to S.O. B.C was supported by JSPS Research Fellowships for Young Scientists. The funders had no role in study design, data collection and analysis, decision to publish, or preparation of the manuscript.

**Competing interests:** The authors have declared that no competing interests exist.

oncogenesis pathway among different tumors is crucial for developing effective anti-cancer strategies. Here, using fruit fly *Drosophila* as a model organism, we searched for a common oncogenesis pathway between two malignant tumor models with different oncogenic alterations, namely oncogenic Ras activation with cell polarity defects ($Ras^{V12}/scrib^{-/-}$) and oncogenic microRNA bantam overexpression with endocytic defects (bantam/$rab5^{-/-}$). Transcriptome analyses followed by an RNAi screen identified Juvenile hormone Inducible-21 (JhI-21), a *Drosophila* homolog of the L-amino acid transporter 1 (LAT1), as essential for growth and malignancy of these tumor models. Similar to mammalian cancers, JhI-21 was upregulated in these tumors and promoted tumor growth and progression by activating the mTOR-S6 pathway. Administration of LAT1 inhibitors to flies significantly suppressed growth of $Ras^{V12}/scrib^{-/-}$, but not bantam/$rab5^{-/-}$ tumors. Our analyses revealed that bantam induces drug resistance against LAT1 inhibitors via downregulation of the TMEM135-like gene *CG31157*. Our observations establish a powerful genetic model for studying cancer progression and drug resistance in *Drosophila*.

## Introduction

Cancer development is achieved by the accumulation of oncogenic mutations that promote cell proliferation, survival, invasion, and metastasis. Mutations that drive tumor growth and malignancy are different between tumors and thus identification of a common oncogenesis pathway among tumors with different oncogenic alterations is crucial for establishing effective anti-cancer strategies.

In recent years, amino acid transporter, especially L-amino acid transporter 1 (LAT1), has been attracting attention as a potential therapeutic target for cancer. LAT1 is a plasma membrane transporter for branched-chain amino acids (BCAAs) such as leucine and isoleucine[1], thereby promoting tumor growth by activating mTOR signaling[2]. LAT1 acts as a protein complex composed of LAT1 covalently bound to 4F2 Heavy Chain Antigen (CD98/SLC3A2) [3]. CD98 promotes LAT1 protein stability and mediates the translocation of LAT1 to the cell membrane[3]. Studies in mammalian cells have shown that LAT1 is upregulated in neuroblastoma and Burkitt's lymphoma cells via Myc[4] and in breast cancer cells via aryl hydrocarbon receptor (AHR)-mediated signaling[5]. Given that LAT1 expression is elevated in a variety of cancers, it is thought to be an ideal therapeutic target as a component of a common oncogenesis pathway and is thus currently under clinical trial in cancer patients[2,6]. However, genetic complexity and heterogenous nature of cancer have hindered progress in understanding the mechanism of the common genetic pathway of tumor growth and malignancy in mammalian systems.

The genetic mosaic technique available in *Drosophila* provides an ideal model system to study tumor growth and progression with genetically traceable oncogenic mutations. Indeed, previous studies in *Drosophila* imaginal epithelium have identified critical mechanisms by which accumulation of distinct oncogenic alterations drives tumor malignancy. For instance, clones of Ras-activated benign tumors are transformed into malignant tumors when simultaneously mutated for an apicobasal polarity gene such as *scribble* (*scrib*)[7,8]. In addition, clones of cells activating Ras and Src signaling develop into invasive tumors under high-sugar diet condition[9].

Here, through a transcriptome analysis combined with *Drosophila* genetics, we searched for a common pathway of oncogenesis among different types of *Drosophila* malignant tumors. By an unbiased transcriptome analysis, we found that LAT1 expression was commonly elevated

in *Drosophila* malignant tumors with different oncogenic mutations. Genetic or pharmacological inhibition of LAT1 significantly blocked tumor growth and malignancy in these tumors. Our findings unveil an evolutionarily conserved role of LAT1 induction in tumor progression and provide a novel genetic model for analyzing cancer progression and drug resistance.

## Results

### JhI-21/LAT1 is required for tumor growth and invasion in *Drosophila*

In *Drosophila* imaginal epithelia, clones of cells overexpressing oncogenic Ras$^{V12}$ with simultaneous mutations in apico-basal polarity genes such as *scribble* (*scrib*) or *discs large* (*dlg*) result in tumorous overgrowth and metastatic behavior, the best-characterized model of *Drosophila* malignant tumors[7,8] (Fig 1b, compare to Fig 1a). To study a common pathway of oncogenesis among different types of malignant tumors, we first tried to establish another model of *Drosophila* malignant tumors using different oncogenic mutations. As a result, we found that loss-of-function mutations in a tumor-suppressor gene *rab5*[10], a small GTPase essential for generating early endosomes[11], in clones of cells overexpressing a pro-growth microRNA bantam (bantam/*rab5*$^{-/-}$ cells), a target of the Hippo pathway effector Yorkie (Yki), in the eye discs resulted in drastic tumor growth and malignant invasion to adjacent organ ventral nerve cord (VNC) (Figs 1c and S1e, quantified in S1f Fig). Notably, overexpression of bantam alone or *rab5* mutation alone caused neither tumor growth (S1a and S1c Fig, quantified in S1f Fig) nor metastatic invasion (S1b and S1d Fig), suggesting an oncogenic cooperation between these alterations. Thus, we established a new model of *Drosophila* malignant tumors caused by bantam overexpression with *rab5* defect.

Using these two malignant tumor models with distinct oncogenic mutations, we performed RNA sequence (RNA-seq) analyses from GFP-labeled fluorescence-activated cell sorting (FACS)-sorted Ras$^{V12}$/*scrib*$^{-/-}$ or bantam/*rab5*$^{-/-}$ cells compared to GFP-labeled wild-type cells (Fig 1d). We identified 4,553 and 2,471 genes that are significantly upregulated or downregulated in Ras$^{V12}$/*scrib*$^{-/-}$ and bantam/*rab5*$^{-/-}$ cells, respectively, and 1,734 genes that are commonly upregulated (1,028 genes) or downregulated (706 genes) in both tumors compared to wild-type cells (Fig 1e and S1 Table, false discovery rate (FDR)< 0.05). Among 1,028 commonly upregulated genes, we found JNK signaling targets such as *mmp1*[12] and *puckered* (*puc*)[13], JAK/STAT signaling targets such as *upd1*, *upd2*, and *upd3*[14], and Yki targets such as *expanded*, *cycE*, and *crb*[15] (Fig 1f and S1 Table), which are consistent with previous reports for the upregulated genes in Ras$^{V12}$/*scrib*$^{-/-}$ tumors[16–18]. These data validate our experimental conditions and suggest that these malignant tumors with distinct mutant origins share common downstream signaling to induce tumor growth and invasion.

We next performed an *RNAi* screen to identify common essential components for driving tumor growth and invasion in commonly upregulated genes. Among 75 upregulated genes (S1 Table), we found that knocking down of *JhI-21* (a LAT1 homolog[19]), *minidiscs* (*mnd*, a y +LAT1 homolog), *cacophony* (*cac*, a voltage-gated calcium channel) or *Calmodulin* (*Cam*) significantly suppressed Ras$^{V12}$/*scrib*$^{-/-}$ or Ras$^{V12}$/*dlg*$^{-/-}$ tumor growth (Fig 1g, quantified in Fig 1o; S1g Fig). The strongest effect was observed by *JhI-21* knockdown (using two different RNAi lines HMS02271 and KK112996; Fig 1o). Although the stronger *JhI-21* knockdown (HMS02271) suppressed growth of wild-type clones (Fig 1m' and 1n', quantified in Fig 1o), which suggests that JhI-21 is essential for normal tissue growth, it completely abolished tumor growth and invasion of Ras$^{V12}$/*scrib*$^{-/-}$ or bantam/*rab5*$^{-/-}$ tumors (Fig 1j', compare to Fig 1g', quantified in Fig 1o; Fig 1l, compare to Fig 1b; S1j Fig, compared to S1h Fig, quantified in S1l Fig) and rescued lethality of animals bearing these tumors (Fig 1k, compare to Fig 1h and 1i; S1k Fig, compared to S1i Fig). Together, these data indicate that Ras$^{V12}$/*scrib*$^{-/-}$ and bantam/

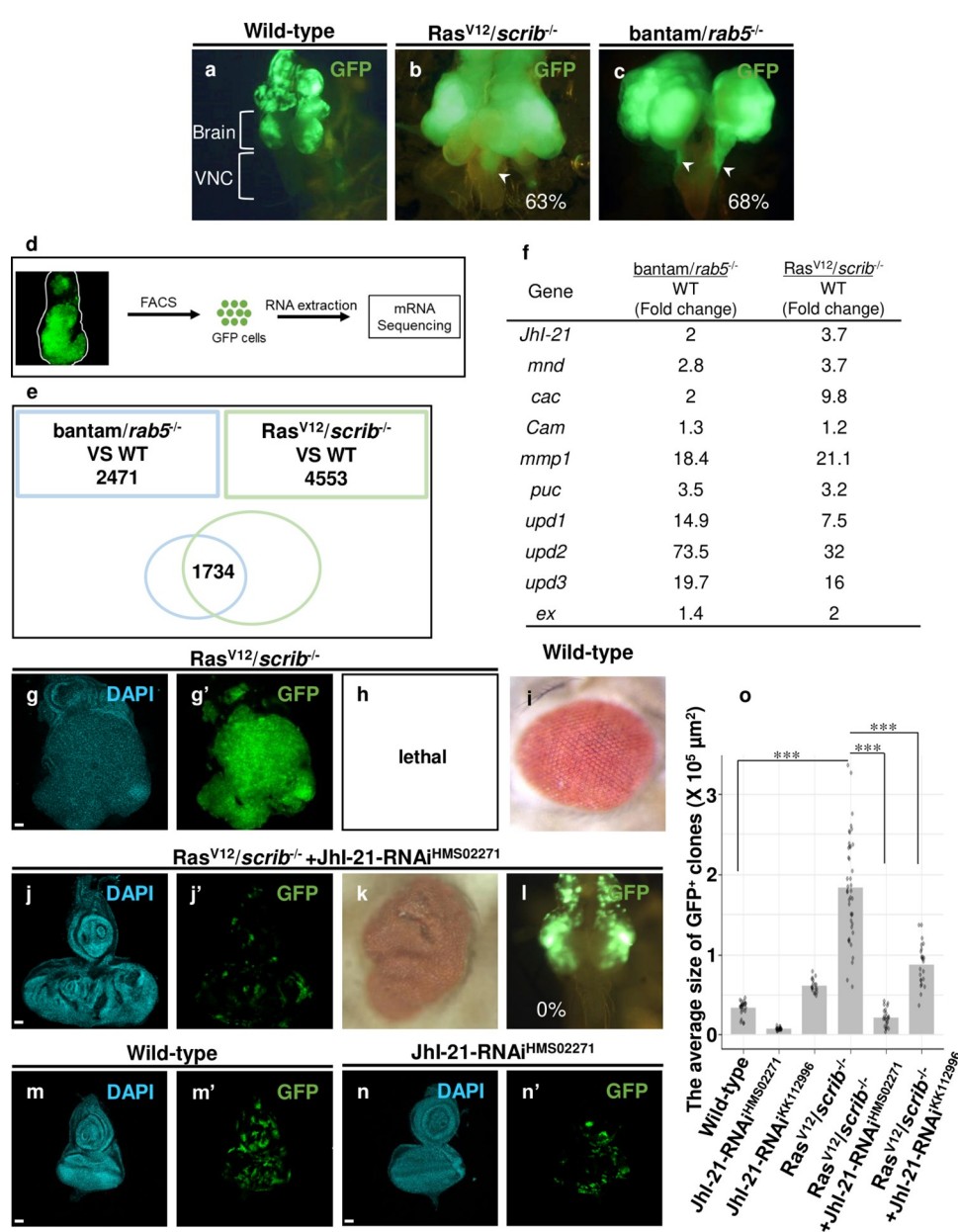

**Fig 1. L-amino acid transporter is required for tumor growth and progression.** (a, b, c and l) Images of cephalic complexes, which include brain and the ventral nerve cord (VNC). Wild-type (a) and Ras$^{V12}$/*scrib*$^{-/-}$ or bantam/*rab5*$^{-/-}$ tumors showing VNC invasion at 9 days after egg laying are shown (arrowheads, 63% of animals show VNC invasion, n = 16 (b); 68% of animals show VNC invasion, n = 31 (c)). At 7 days after egg laying, Ras$^{V12}$/*scrib*$^{-/-}$ +JhI-21-RNAi flies did not show VNC invasion (0% of animals, n = 19 (l)). (d) Transcriptional profiling of wild-type cells (WT) or tumor cells by RNA-seq was performed with mRNA isolated from FACS-sorted GFP$^+$ cells. (e) Venn diagram of RNA-seq shows the number of genes whose expressions were changed relative to Wild-type (WT) (PDR<0.05) in eye/antennal imaginal discs bearing bantam/*rab5*$^{-/-}$ (2,471 genes) or Ras$^{V12}$/*scrib*$^{-/-}$ (4,553 genes). 1,734 genes were overlapped in both bantam/*rab5*$^{-/-}$ and Ras$^{V12}$/*scrib*$^{-/-}$ cells. (f) Representative genes whose expression levels were upregulated in both bantam/*rab5*$^{-/-}$ and Ras$^{V12}$/*scrib*$^{-/-}$ tumor cells compared to WT. (g, h, j, k, l, m and n) Eye-antennal discs bearing GFP-labeled Ras$^{V12}$/*scrib*$^{-/-}$ (g and h), Ras$^{V12}$/*scrib*$^{-/-}$ + JhI-21-RNAi (HMS02271) (j, k and l), Wild-type (m), and JhI-21-RNAi (HMS02271) (n) clones were shown. (h, i and k) Adult eye phenotypes, lethality (h), Wild-type (i) and Ras$^{V12}$/*scrib*$^{-/-}$ + JhI-21-RNAi (HMS02271) (k) are shown. Larvae bearing bantam/*rab5*$^{-/-}$ or Ras$^{V12}$/*scrib*$^{-/-}$ tumors in the eye-antennal discs failed to develop into adult flies. (o) Average sizes of Wild-type (n = 26), JhI-21-RNAi (HMS02271, n = 22), JhI-21-RNAi (KK112996, n = 18), Ras$^{V12}$/*scrib*$^{-/-}$ (n = 41), and Ras$^{V12}$/*scrib*$^{-/-}$ +JhI-21-RNAi (HMS02271, n = 23), RasV12/*scrib*$^{-/-}$ +JhI-21-RNAi (KK112996, n = 24) clones were measured by ImageJ. Scale bars, 50 μm. See S3 Table for details of statistical analyses.

$rab5^{-/-}$ malignant tumors upregulate JhI-21/LAT1, which is essential for their growth and invasion.

## Activation of JNK and Yki upregulates JhI-21

LAT1 is a plasma membrane transporter for BCAAs such as leucine and isoleucine and is often upregulated in tumor cells, thereby promoting tumor growth[6]. However, the mechanism by which LAT1 is upregulated in various cancer cells is still unclear. We thus investigated the mechanism by which JhI-21 is upregulated in $Ras^{V12}$/$scrib^{-/-}$ or bantam/$rab5^{-/-}$ tumor cells. Consistent with the RNA-seq data, immunostaining for JhI-21 protein in the eye-antennal discs showed upregulation of JhI-21 in $Ras^{V12}$/$scrib^{-/-}$. (Fig 2a, quantified in Fig 2g) or bantam/$rab5^{-/-}$ (S2b Fig) clones. In addition, qRT-PCR analysis confirmed that the level of JhI-21 mRNA expression was significantly upregulated in $Ras^{V12}$/$scrib^{-/-}$ cells compared to wild-type (S2a Fig). It has been reported that activation of JNK signaling is essential for overgrowth and invasion of $Ras^{V12}$/$scrib^{-/-}$ or $Ras^{V12}$/$dlg^{-/-}$ tumors[16]. Indeed, JNK activity was elevated in both $Ras^{V12}$/$scrib^{-/-}$ and bantam/$rab5^{-/-}$ tumors as visualized by the immunostaining of JNK target MMP1 (S2c and S2d Fig) as well as by the RNA-seq data showing upregulation of JNK targets *mmp1* and *puc* (Fig 1e). Strikingly, blocking JNK signaling by overexpression of a dominant-negative form of the *Drosophila* JNK Bsk (Bsk^DN) in $Ras^{V12}$/$scrib^{-/-}$ or bantam/$rab5^{-/-}$ tumors abolished JhI-21 induction in these tumors and blocked their growth (Fig 2b, quantified in Fig 2g; S2e Fig). This indicates that these tumors upregulate JhI-21 expression via JNK activation. However, JNK activation alone by overexpressing Eiger (a tumor necrosis factor (TNF) homolog that activates JNK signaling[20,21]) did not induce JhI-21 expression (Fig 2c, quantified in Fig 2g), suggesting that additional factor is required for JhI-21 induction in conjunction with JNK activation. Interestingly, we found that overexpression of a Hippo pathway component Warts (Wts, a Lats homolog that suppresses Yki/YAP activity[22]) in $Ras^{V12}$/$scrib^{-/-}$ tumors abolished the upregulation of JhI-21 (Fig 2d, quantified in Fig 2g), suggesting that Yki activity is also required for JhI-21 induction. Indeed, Yki activation was observed in both $Ras^{V12}$/$scrib^{-/-}$ and bantam/$rab5^{-/-}$ tumors, as visualized by Yki activity reporter *expanded* (*ex*)-*lacZ* or *four-jointed* (*fj*)-*lacZ* (Figs 2f and S2g) as well as by the RNA-seq data showing upregulation of Yki targets *expanded*, *cycE*, *crb*, and *upd1* (Fig 1e and S1 Table). However, Yki activation alone by overexpressing an activated form of Yki (Yki^S168A) did not cause JhI-21 induction (Fig 2e, quantified in Fig 2g). Significantly, we found that co-activation of JNK and Yki caused JhI-21 induction (Fig 2f, quantified in Fig 2g). These data indicate that activation of JNK and Yki in $Ras^{V12}$/$scrib^{-/-}$ or bantam/$rab5^{-/-}$ tumors cause upregulation of JhI-21 expression.

## JhI-21 promotes mTOR-S6 signaling

We next invested the consequence of JhI-21 induction in tumor clones. JhI-21 and its mammalian homolog LAT1 play roles in uptalking leucine[1,19,23]. It has been reported that leucine uptake into the cell activates an evolutionarily conserved mechanistic target of rapamycin (mTOR) signaling in mammalian cells[24]. In the mTOR pathway, mTORC1 (mTOR kinase complex 1) directly phosphorylates S6 kinase 1 (S6K1), which phosphorylates ribosomal protein S6 (RpS6) and thus promotes protein translation[25]. It has been shown that LAT1 contributes to tumor growth by activating mTOR signaling[2]. We found that phosphorylation of RpS6 was significantly elevated in $Ras^{V12}$/$scrib^{-/-}$ and bantam/$rab5^{-/-}$ tumors (Fig 3a and 3b, quantified in Fig 3f), while RpS6 phosphorylation was not detected in $scrib^{-/-}$, $Ras^{V12}$, or $rab5^{-/-}$ cells (S3b–S3d Fig, compare to S3a Fig, quantified in S3g Fig). On the other hand, moderate levels of RpS6 phosphorylation were observed in a part of bantam-expressing cells (S3e Fig, quantified in S3g Fig). Crucially, knockdown of JhI-21 strongly suppressed phosphorylation of

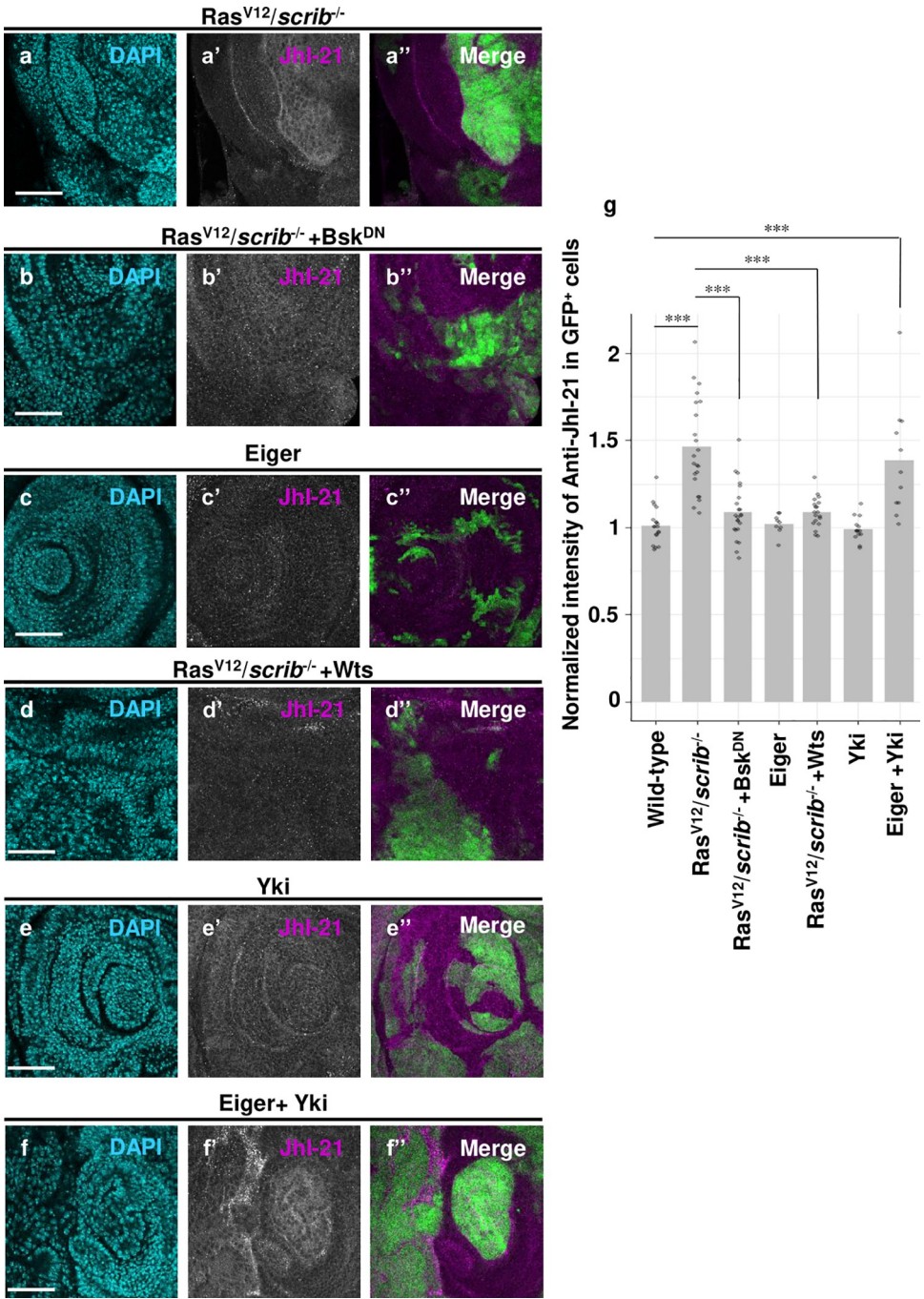

**Fig 2. Co-activation of JNK and Yki upregulates JhI-21.** (a, b, c, d, e and f) GFP-labeled Ras$^{V12}$/*scrib*$^{-/-}$ (a), Ras$^{V12}$/*scrib*$^{-/-}$ +Bsk$^{DN}$ (b), Eiger (c), Ras$^{V12}$/*scrib*$^{-/-}$ +Wts (d), Yki (e), and Eiger +Yki (f) clones were induced in eye-antennal discs and were stained with anti-JhI-21 antibody. (g) The averaged normalized intensity of anti-JhI-21 staining in GFP$^+$ cells in each genotype: Wild-type (n = 19), Ras$^{V12}$/*scrib*$^{-/-}$ (n = 22), Ras$^{V12}$/*scrib*$^{-/-}$ +Bsk$^{DN}$ (n = 23), Eiger (n = 9), Ras$^{V12}$/*scrib*$^{-/-}$ +Wts (n = 19), Yki (n = 14), Eiger +Yki (n = 11). The intensity was measured by ImageJ. Cell nuclei were stained with DAPI. Scale bars, 50 μm. See S3 Table for details of statistical analyses.

RpS6 in both Ras$^{V12}$/*scrib*$^{-/-}$ and bantam/*rab5*$^{-/-}$ tumors (Fig 3c and 3d, quantified in Fig 3f). Western blot analysis for the phosphorylation of the eukaryotic initiation factor 4E binding protein (4EBP), an indicator of mTOR activation[26], confirmed that JhI-21 knockdown

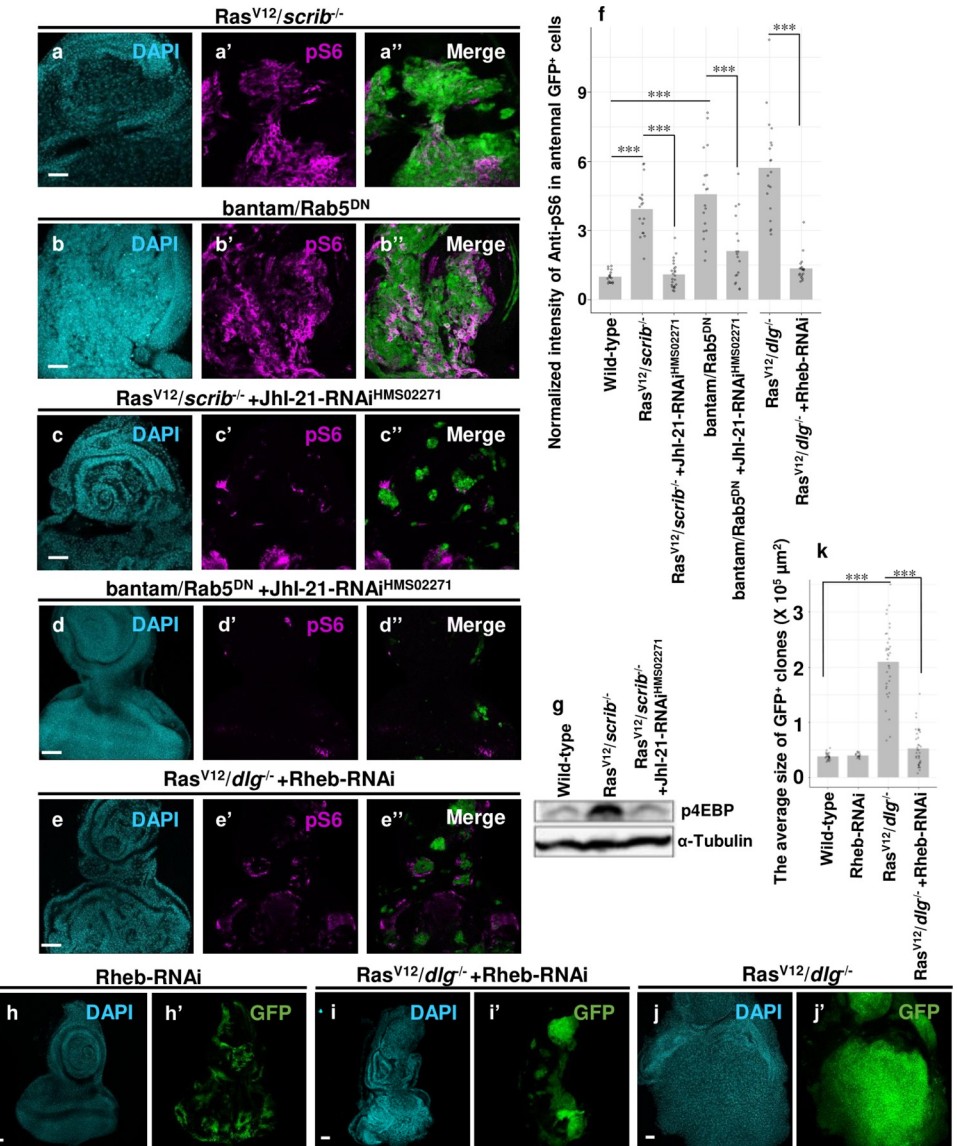

**Fig 3. JhI-21 contributes to tumor growth by activating the mTOR-S6 pathway.** (a, b, c, d and e) GFP-labeled Ras$^{V12}$/*scrib*$^{-/-}$ (a), bantam/Rab5$^{DN}$ (b), Ras$^{V12}$/*scrib*$^{-/-}$ +JhI-21-RNAi (HMS02271) (c), bantam/Rab5$^{DN}$ +JhI-21-RNAi (HMS02271) (d), and Ras$^{V12}$/*dlg*$^{-/-}$ +Rheb-RNAi (e) clones were induced in eye-antennal discs and were stained with anti-pS6 antibody. (f) The averaged normalized intensity of anti-pS6 staining in antennal GFP$^+$ cells in each genotype: Wild-type (n = 17), Ras$^{V12}$/*scrib*$^{-/-}$ (n = 18), Ras$^{V12}$/*scrib*$^{-/-}$ +JhI-21-RNAi (HMS02271) (n = 23), bantam/Rab5$^{DN}$ (n = 18), bantam/Rab5$^{DN}$ + JhI-21-RNAi (HMS02271) (n = 18), Ras$^{V12}$/*dlg*$^{-/-}$ (n = 19), and Ras$^{V12}$/*dlg*$^{-/-}$ +Rheb-RNAi (n = 19). The intensity was measured by ImageJ. (g) Western blot analysis of p4EBP from Wild-type, Ras$^{V12}$/*scrib*$^{-/-}$ or Ras$^{V12}$/*scrib*$^{-/-}$ +JhI-21-RNAi (HMS02271) eye-antennal discs. (h, i and j) GFP-labeled Rheb-RNAi (h), Ras$^{V12}$/*dlg*$^{-/-}$ +Rheb-RNAi (i), and Ras$^{V12}$/*dlg*$^{-/-}$ (j) clones were induced in eye-antennal discs. (k) Average sizes of Wild-type (n = 28), Rheb-RNAi (n = 18), Ras$^{V12}$/*dlg*$^{-/-}$ (n = 35), and Ras$^{V12}$/*dlg*$^{-/-}$ +Rheb-RNAi (n = 38) clones were measured by ImageJ. Cell nuclei were stained with DAPI. Scale bars, 50 μm. See S3 Table for details of statistical analyses.

blocked mTOR signaling activation in Ras$^{V12}$/*scrib*$^{-/-}$ tumors (Fig 3g, quantified in S3h Fig). In addition, blocking mTOR signaling by knocking down of an upstream regulator of mTOR signaling Rheb[24] significantly suppressed RpS6 phosphorylation in Ras$^{V12}$/*dlg*$^{-/-}$ tumors (Fig 3e, compare to S3f Fig, quantified in Fig 3f) and tumor growth (Fig 3i, compare to Fig 3j, quantified in Fig 3k), while Rheb knockdown alone did not affect tissue growth (Fig 3h, quantified

in Fig 3k). Together, these data suggest that JhI-21 upregulation promotes tumor growth by activating mTOR-S6 signaling in *Drosophila*.

## Administration of LAT1 inhibitors reduces growth of Ras$^{V12}$/*scrib*$^{-/-}$ tumors

We next examined whether pharmacological inhibition of JhI-21 activity could suppress growth of malignant tumors in *Drosophila*. It is known that LAT1 inhibitors, 2-amino-2-Nor-bornanecarboxylic Acid (BCH) and KYT0353 (JPH203), suppress the activity of LAT1 in mammalian cells[6]. KYT0353 is currently being evaluated in a Phase 2 clinical trial in patients with advanced biliary tract cancers (UMIN Clinical Trials Registry UMIN000034080). Notably, we found that feeding BCH or KYT0353 to larvae bearing Ras$^{V12}$/*scrib*$^{-/-}$ tumors in the eye-antennal discs significantly reduced tumor growth, while these drugs did not affect growth of wild-type clones (Figs 4a, 4b, S4a and S4b). Furthermore, BCH treatment significantly suppressed mTOR signaling activity in Ras$^{V12}$/*scrib*$^{-/-}$ tumors (S4c Fig). These data indicate that pharmacological inhibition of JhI-21 activity suppresses growth of Ras$^{V12}$/*scrib*$^{-/-}$ malignant tumors by downregulating mTOR signaling.

## MicroRNA bantam renders malignant tumors resistant to LAT1 inhibitors

To our surprise, BCH and KYT0353 did not suppress growth of bantam/*rab5*$^{-/-}$ tumors (Fig 4a and 4b). Consistent with this result, BCH treatment did not suppress mTOR signaling activity in bantam/*rab5*$^{-/-}$ tumors (S4c Fig). Notably, overexpression of bantam in Ras$^{V12}$/*dlg*$^{-/-}$ tumors abolished the suppressive effect of LAT1 inhibitors on their growth (Figs 4c, 4d, S4d and S4e). These data suggest that bantam renders malignant tumors resistant to LAT1 inhibitory drugs.

To identify gene(s) responsible for the drug resistance against LAT1 inhibitors upon bantam expression, we performed an RNA-seq analysis of FACS-sorted bantam-overexpressing cells compared to wild-type cells in the eye-antennal discs. Expression levels of 42 genes were significantly altered in bantam-overexpressing cells compared to wild-type cells, and among these 10 genes were commonly upregulated (3 genes) or downregulated (7 genes) in both bantam cells and bantam/*rab5*$^{-/-}$ tumors but not in Ras$^{V12}$/*scrib*$^{-/-}$ tumors (Fig 4e and S2 Table, false discovery rate (FDR)<0.05). Crucially, we found that knockdown of *CG31157* (GD3640), one of the commonly downregulated 7 genes encoding a TMEM135-like protein, in Ras$^{V12}$/*dlg*$^{-/-}$ tumors abrogated tumor-suppressive effect of BCH (Fig 4g, compare to Fig 4f, quantified in Fig 4i), while knockdown of *CG31157* (GD3640) on its own did not reduce Ras$^{V12}$/*dlg*$^{-/-}$ tumor burden or wild-type clone size (Figs 4h and S4f, quantified in Figs 4i and S4g). A similar result was obtained by using a different RNAi line for *CG31157* (KK111271) (S4h Fig). Furthermore, bantam/*rab5*$^{-/-}$ tumors overexpressing CG31157 transgene became sensitive to BCH treatment (Fig 4j), indicating a critical role of CG31157 in drug responsiveness. Notably, *CG31157* expression was significantly upregulated in Ras$^{V12}$/*scrib*$^{-/-}$ tumors (~1.7 fold, S2 Table), suggesting that *CG31157* expression is critical for cells to acquire sensitivity to LAT1 inhibitors. These data suggest that bantam renders malignant tumors resistant to LAT1 inhibitory drugs via downregulation of TMEM135-like gene *CG31157*.

## Discussion

Our genetic study using *Drosophila* tumor models revealed that activation of JNK and Yki drives tumor growth and malignancy by inducing JhI-21, a fly homolog of LAT1 (Fig 5). It has previously been shown that JhI-21 acts as an amino acid transporter that uptakes leucine into insulin producing cells (IPCs) in *Drosophila* larva and is required for leucine-dependent secretion of *Drosophila* insulin-like peptide 2 (Dilp2) from IPCs[23]. In this study, we found that

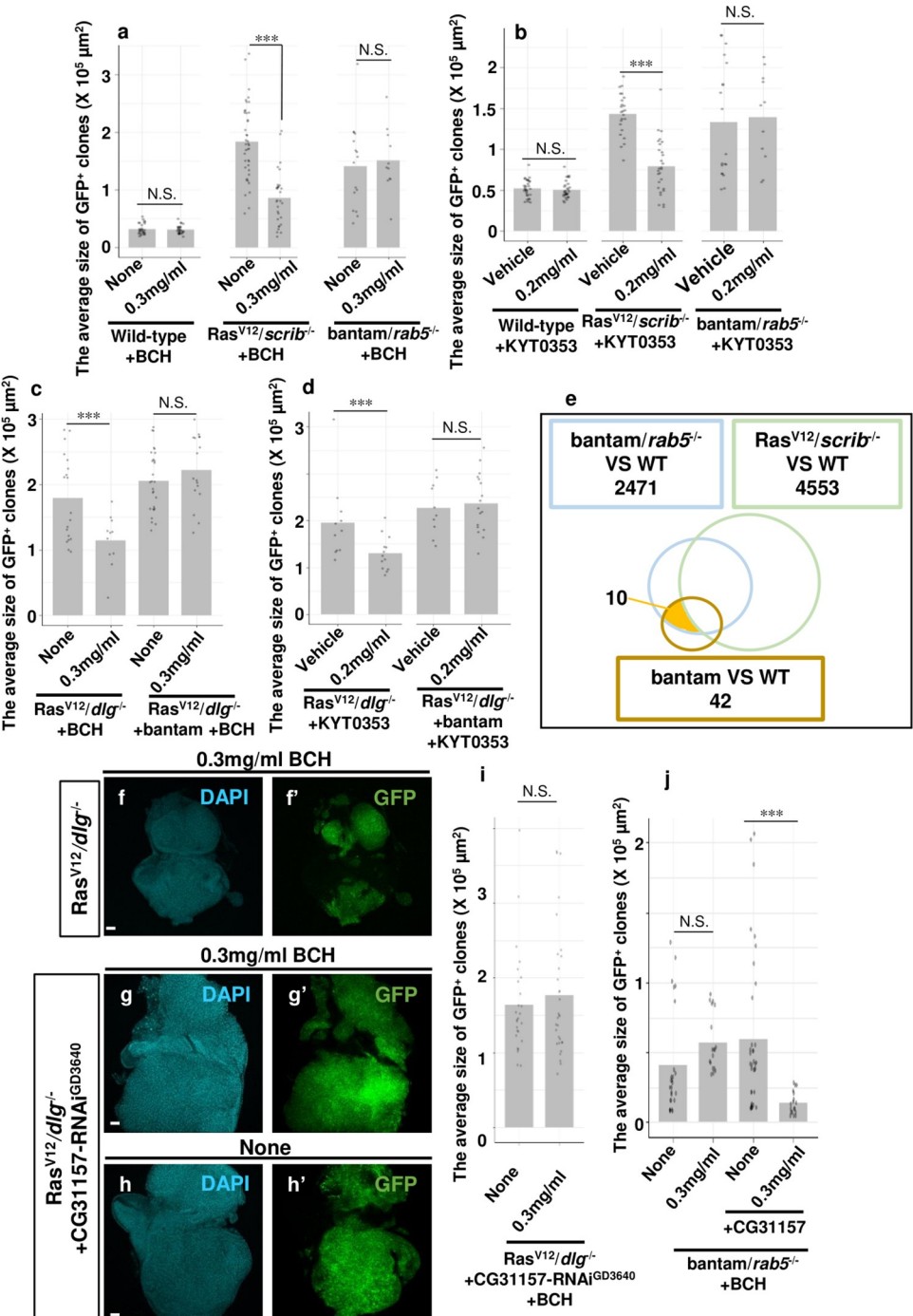

**Fig 4. Tumor growth is attenuated by LAT1 inhibitors and bantam expression induces drug resistance.** (a, b, c and d) Quantification of average sizes of GFP+ clones of Wild-type, Ras$^{V12}$/scrib$^{-/-}$ (day7) (a and b), bantam/rab5$^{-/-}$ (day9) (a and b), Ras$^{V12}$/dlg$^{-/-}$ (day7) (c and d), and Ras$^{V12}$/dlg$^{-/-}$ +bantam (day6) (c and d) after larval administration of Vehicle, BCH or KYT0353. (a) Wild-type (n = 27), Wild-type +BCH (n = 28), Ras$^{V12}$/scrib$^{-/-}$ (n = 41), Ras$^{V12}$/scrib$^{-/-}$ +BCH (n = 30), bantam/rab5$^{-/-}$ (n = 49), bantam/rab5$^{-/-}$ +BCH (n = 40). (b) Wild-type +Vehicle (n = 30), Wild-type +KYT0353 (n = 30), Ras$^{V12}$/scrib$^{-/-}$ +Vehicle (n = 24), Ras$^{V12}$/scrib$^{-/-}$ +KYT0353 (n = 28), bantam/rab5$^{-/-}$ +Vehicle (n = 31), bantam/rab5$^{-/-}$ +KYT0353 (n = 15). (c) Ras$^{V12}$/dlg$^{-/-}$ (n = 19), Ras$^{V12}$/dlg$^{-/-}$ +BCH (n = 11), Ras$^{V12}$/dlg$^{-/-}$ +bantam (n = 24), Ras$^{V12}$/dlg$^{-/-}$ +bantam +BCH (n = 16). (d) Ras$^{V12}$/dlg$^{-/-}$ +Vehicle (n = 11), Ras$^{V12}$/dlg$^{-/-}$ +KYT0353 (n = 13), Ras$^{V12}$/dlg$^{-/-}$ +bantam + vehicle (n = 10), Ras$^{V12}$/dlg$^{-/-}$ +bantam +KYT0353 (n = 17). (e) Venn diagram of RNA-seq showing the number of genes whose expressions were changed relative to Wild-type (WT) (PDR<0.05) in eye/antennal imaginal discs bearing bantam/rab5$^{-/-}$ (2,471 genes), Ras$^{V12}$/scrib$^{-/-}$ (4,553 genes) or bantam (42 genes).

(f, g and h) Eye-antennal discs bearing Ras$^{V12}$/$dlg^{-/-}$ (f) or Ras$^{V12}$/$dlg^{-/-}$ + CG31157-RNAi (GD3640) (g and h) clones in larvae after feeding with (f and g) or without BCH (h). (i) Quantification of average sizes of GFP$^+$ clones of Ras$^{V12}$/$dlg^{-/-}$ + CG31157-RNAi (GD3640) (day7) in larvae after feeding with or without BCH. Ras$^{V12}$/$dlg^{-/-}$ + CG31157-RNAi (GD3640) (n = 26), Ras$^{V12}$/$dlg^{-/-}$ + CG31157-RNAi (GD3640) +BCH (n = 31). (j) Quantification of average sizes of GFP$^+$ clones of bantam/$rab5^{-/-}$ or bantam/$rab5^{-/-}$ + CG31157 (day7) in larvae after feeding with or without BCH. bantam/$rab5^{-/-}$ (n = 26), bantam/$rab5^{-/-}$ +BCH (n = 18), bantam/$rab5^{-/-}$ +CG31157 (n = 36), bantam/$rab5^{-/-}$ +CG31157 +BCH (n = 18). The average sizes of clones were measured by ImageJ. See S3 Table for details of statistical analyses.

JhI-21 is commonly upregulated in *Drosophila* malignant tumors and contributes to tumor growth and malignancy via activation of the mTOR-S6 pathway. Similar to mammalian systems, knockdown or pharmacological inhibition of JhI-21/LAT1 significantly reduced growth of malignant tumors. We also found that *mnd* was commonly upregulated in Ras$^{V12}$/$scrib^{-/-}$ and bantam/$rab5^{-/-}$ tumors and knockdown of *mnd* suppressed tumor growth. Mnd is an amino acid transporter belonging to LAT1 family, which catalyzes the cross-membrane flux of large neutral amino acids by forming heterodimers with CD98[19,27]. Thus, our data suggest that similar to mammalian cancers, elevation of LAT1 activity is critical for tumor growth and progression in *Drosophila*. This indicates that future studies on the common oncogenesis pathway in *Drosophila* could provide novel therapeutic strategies against human cancer.

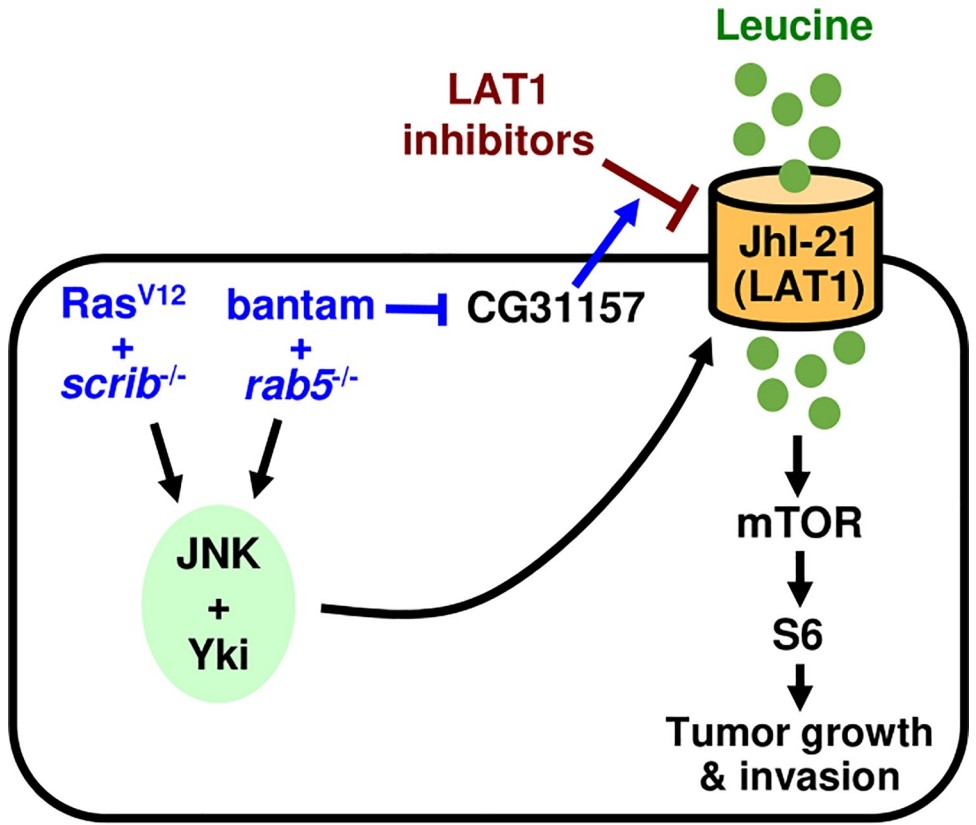

**Fig 5. A model for the induction of JhI-21/LAT1 promoting tumor progression.** Elevated activity of JNK and Yki in *Drosophila* tumors drives tumor growth and invasion by inducing JhI-21, which activates the mTOR-S6 pathway. Elevation of microRNA bantam renders tumors resistant to LAT1 inhibitors by downregulating a TMEM135-like gene CG31157.

Indeed, our data intriguingly show that overexpression of a pro-growth microRNA bantam rendered malignant tumors resistant to LAT1 inhibitory drugs. Our data suggest that the bantam-induced drug resistance against LAT1 inhibitors is due to downregulation of *CG31157*, which encodes a TMEM135-like protein that may regulate mitochondrial morphology and dynamics[28]. Thus, our observations not only show an evolutionarily conserved role of LAT1 induction in driving tumor malignancy but provide a useful genetic model for studying the mechanism of drug resistance. Notably, human esophageal cancer cells were resistant to KYT0353, in spite that LAT1 is upregulated in these cells[29]. Thus, although the mechanism by which *CG31157* contributes to LAT1 inhibition by BCH and KYT0353 is currently unknown, future studies on the underlying mechanisms could contribute to improve drug resistance in cancer therapies.

In this study, we found that co-activation of JNK and Yki leads to upregulation of JhI-21 in *Drosophila* imaginal discs. The mechanism by which JNK and Yki induce JhI-21 expression is currently unknown, which should be addressed in the future studies. Nonetheless, JNK activation has been shown to be essential for tumor growth and invasion in *Drosophila* malignant tumors[16] and JNK activation has long been implicated in tumor growth and progression in mammalian systems[30–32], underscoring the critical role of JNK in tumor progression. Indeed, hyperactivation of JNK signaling, as well as elevated YAP activity, have been reported in many human cancers[24,32]. Given that signaling molecules identified in this study are all conserved in humans, similar tumor progression mechanism via JNK and YAP-mediated LAT1 induction could regulate human cancers.

## Materials and methods

### *Drosophila* strains and genetics

Fly stocks were cultured at room temperature or 25˚C on standard fly food. Fluorescently-labeled mitotic clones[33,34] were produced in larval imaginal discs using the following strains: *eyFLP1; TubGal80, FRT40A; Act> y+ >Gal4, UAS-GFP* (40A tester); *eyFLP1; Act> y+ >Gal4, UAS-GFP; FRT82B, TubGal80* (82B tester); *FRT19A, Tub-Gal80; eyFRP5, Act>y+> Gal4, UAS-GFP; sb/TM6B* (19A tester). Additional strains used are as follows: UAS-Ras$^{V12}$ (BL4847, BL5788), UAS-bantam (BL60672), UAS-JhI-21-RNAi (BL41706), UAS-Rab5$^{DN}$ (BL42704), UAS-Yki$^{S168A}$ (BL28818), UAS-Rheb-RNAi (BL33966), *rab5*$^{LL00467}$ (DGRC, Kyoto Institute of Technology), UAS-JhI-21-RNAi (VDRC108509), UAS-CG31157-RNAi (VDRC30296), UAS-Bsk$^{DN}$ (T. Adachi-Yamada), *scrib*$^{1}$ (D. Bilder)[35], UAS-Eiger$^{W}$, UAS-Eiger$^{12}$ (M. Miura), *dlg*$^{m52}$ (N. Perrimon), UAS-Wts (T. Xu).

CG31157 was PCR-amplified from cDNA of w$^{1118}$ adult flies using primers designed to append restriction sites for enzymes EcoR1 and Xbal to the 5' and 3' end of the product. Sequence of the products were confirmed and then the product was cloned into the pUAST vector. Transgenic flies were generated by WellGenetics.

### Immunohistochemistry

Larval tissues were stained with standard immunohistochemical procedures using rabbit anti-JhI-21[23] (1:100; Dr. Grosjean; in Can Get Signal Immunoreaction Enhancer Solution A), rabbit anti-Phospho-S6[36] (1:400; J. Chung). Secondary antibodies used: anti-rabbit Alexa-fluor 546 or 647 (1:250). Samples were mounted with DAPI-containing SlowFade Gold Anti-fade Reagent (Molecular Probes). Images were taken with a Leica SP5. Clone size and intensity of anti-JhI-21 or anti-pS6 in GFP$^{+}$ cells were measured using ImageJ. Graphs were drawn by using ggplot2 package of R.

## RNA-seq data analysis

Reads were trimmed to 75 nucleotides length by fastx_trimmer in FASTX Toolkit (v0.0.14), and further quality-filtered by trim_galore (v0.5.0) with default setting to remove the adaptor sequence and the low quality reads. The reads passing filters were mapped to the *Drosophila melanogaster* Ensembl BDGP6 obtained from illumine iGenomes by STAR (v2.7.0e)[37]. >80% of reads were uniquely mapped for each experiment. The number of reads that map to each gene was counted by htseq-count (HTSeq v0.11.2) with -s reverse option[38]. Normalization was carried out using calcNormFactors function (edgeR)[39,40]. Differentially expressed genes were identified using glmQLFit and glmQLFTest function in edgeR at a FDR threshold 0.05. R version 3.6.0, edgeR_3.26.1 limma_3.40.0.

## Chemicals

BCH (8.0 mg/ml in water; Cayman Chemical), KYT 0353 (JPH203, 7.4 mg/ml in 40% (w/v) Captisol (CyDex Pharmaceuticals); Tocris) was supplemented in fly food. Flies were cultured with drug-containing food for 1 days and then grown in the vial with normal food.

## Western blot analysis

The third instar larvae were dissected in 1x PBS and eye-antennal discs (35~40) were put into 30 ul 2x sample buffer (4% SDS, 20% glycerol, 0.125M Tris-HCl (pH 6.8), 0.005% BPB, 12% 2-mercaptoethanol). After homogenization at 100˚C for 5 min, samples were centrifuged at 13,000 rpm for 10 min. Lysates for the Western blot analysis were made by the supernatant. Primary antibodies used are as follows: p-4E-BP1 (Thr37/46, 1:500 in 5%BSA/TBST (0.1% Tween20 in TBS), rabbit mAb, cell signaling #2855S), monoclonal anti-α-Tubulin (1:2000 in Can get signal solution1, SIGMA T5168-2ML). Secondary antibodies used are as follows: anti-mouse IgG, HRP-linked antibody (1:10000 in Can get signal solution2, cell signaling #7076S). Images were taken with a LAS-4000 GE.

## Statistical analysis

The areas of GFP-labeled clones were measured by ImageJ. Statistical analysis was performed using Prims9. All experiments shown in the same graph were done at the same time. N.S P (>0.12), * P(0.033), ** P(0.002) and *** P<0.001. All statistical data were summarized in S3 Table.

## Supporting information

**S1 Fig. An RNAi screen for the common pathway of tumorigenesis.** (a, c, e, h, i, j and k) Eye-antennal disc bearing GFP-labeled *rab5*[-/-] (a), bantam (c), bantam/*rab5*[-/-] (e), bantam/ Rab5[DN] (h and i), and bantam/Rab5[DN] +JhI-21-RNAi (HMS02271) (j and k) clones are shown. Cell nuclei were stained with DAPI. (b and d) Images of cephalic complexes, which include brain and the ventral nerve cord (VNC). (f) The average size of Wild-type (n = 26), *rab5*[-/-] (n = 40), bantam/*rab5*[-/-] (n = 33), and bantam (n = 18) clones were measured by ImageJ. (g) The average size of clones of GFP-labeled Ras[V12]/*dlg*[-/-] (n = 35), Ras[V12]/*dlg*[-/-] +cac-RNAi (n = 29), Ras[V12]/*dlg*[-/-] +Cam-RNAi (n = 32), Ras[V12]/*scrib*[-/-] (n = 41), and Ras[V12]/*scrib*[-/-] +mnd-RNAi (n = 25) clones were measured by ImageJ. (l) The average size of Wild-type (n = 26), bantam/Rab5[DN] (n = 23), and bantam/Rab5[DN] +JhI-21RNAi (HMS02271) (n = 25) clones were measured by ImageJ. Scale bars, 50 μm. See S3 Table for details of statistical analyses.
(TIF)

**S2 Fig. Upregulation of JhI-21 is required for tumor growth.** (a) Quantitative RT-PCR revealed equal JhI-21 mRNA expression levels among genotypes. Statistics analyses see S3 Table. (b, d and g) GFP-labeled bantam/$rab5^{-/-}$ clones were induced in eye-antennal discs and were stained with anti-JhI-21 antibody (b), anti-MMP1 (a JNK target) antibody (d) or anti-β-galactosidase antibody to detect Yki-target *fj-lacZ* expression (g). (c and f) GFP-labeled Ras$^{V12}$/$scrib^{-/-}$ clones were induced in eye-antennal discs and were stained with anti-MMP1 antibody (c) or anti-β-galactosidase antibody to detect Yki-target *ex-lacZ* expression (f). (e) GFP-labeled bantam/$rab5^{-/-}$ + Bsk$^{DN}$ clones were induced in eye-antennal discs and were stained with anti-JhI-21 antibody. Cell nuclei were stained with DAPI. Scale bars, 50 μm. (TIF)

**S3 Fig. mTOR-S6 signaling upregulated in tumor cells.** (a, b, c, d e and f) GFP-labeled Wild-type (a), $scrib^{-/-}$ (b), Ras$^{V12}$ (c), $rab5^{-/-}$ (d), bantam (e), and Ras$^{V12}$/$dlg^{-/-}$ (f) clones were induced in eye-antennal disc and were stained with anti-pS6 antibody. (g) The averaged normalized intensity of anti-pS6 in antennal GFP+ cells in each genotype: Wild-type (n = 17), $scrib^{-/-}$ (n = 15), Ras$^{V12}$ (n = 10), $rab5^{-/-}$ (n = 18), bantam (n = 7). (h) The average normalized intensity of anti-p4EBP in antennal discs in each genotype: Wild-type (n = 3), Ras$^{V12}$/$scrib^{-/-}$ (n = 3), Ras$^{V12}$/$scrib^{-/-}$ +JhI-21-RNAi (HMS02271) (n = 3) and normalized by α-Tubulin. The intensity was measured by ImageJ. Cell nuclei were stained with DAPI. Scale bars, 50 μm. See S3 Table for details of statistical analyses. (TIF)

**S4 Fig. Effects of LAT1 inhibitors on tumor growth.** (a and b) Quantification of average sizes of GFP$^+$ clones of Ras$^{V12}$/$scrib^{-/-}$ (day7) after larval administration of different dosages BCH (a) or KYT0353 (b). (a) Ras$^{V12}$/$scrib^{-/-}$ (None, n = 41) Ras$^{V12}$/$scrib^{-/-}$ (0.2mg/ml, n = 42) Ras$^{V12}$/$scrib^{-/-}$ (0.3mg/ml, n = 68) Ras$^{V12}$/$scrib^{-/-}$ (0.4mg/ml, n = 17). (b) Ras$^{V12}$/$scrib^{-/-}$ (Vehicle, n = 24) Ras$^{V12}$/$scrib^{-/-}$ (0.1mg/ml, n = 40) Ras$^{V12}$/$scrib^{-/-}$ (0.2mg/ml, n = 28) Ras$^{V12}$/$scrib^{-/-}$ (0.3mg/ml, n = 40). (c) The averaged normalized intensity of anti-pS6 in antennal GFP+ cells in each genotype: Ras$^{V12}$/$scrib^{-/-}$ (n = 22), Ras$^{V12}$/$scrib^{-/-}$ +BCH (0.3mg/ml) (n = 25), bantam/$rab5^{-/-}$ (n = 15), bantam/$rab5^{-/-}$ +BCH (0.3mg/ml) (n = 18). The intensity was measured by ImageJ. (d and e) Images of cephalic complexes, which include brain and the VNC. GFP-labeled Ras$^{V12}$/$dlg^{-/-}$ + bantam clones were induced in eye-antennal disc and the larvae were administrated with Vehicle or KYT0353 (0.2mg/ml) (images are cephalic complexes at day 7). (f) GFP-labeled CG31157-RNAi (GD3640)-expressing clones were induced in eye-antennal disc. Cell nuclei were stained with DAPI. Scale bars, 50 μm. (g) The average size of Wild-type (n = 26) and CG31157-RNAi (GD3640) (n = 11) clones were measured by ImageJ. (h) Quantification of average sizes of GFP$^+$ clones of Ras$^{V12}$/$scrib^{-/-}$ (day7, n = 23) or Ras$^{V12}$/$scrib^{-/-}$ +-CG31157-RNAi (KK111271) (day7, n = 23) after larval administration of BCH. See S3 Table for details of statistical analyses. (TIF)

**S1 Table. RNA-seq data for Ras$^{V12}$/scrib-/- and bantam/$rab5$-/- cell clones mosaically induced in the eye-antennal discs.** (XLSX)

**S2 Table. RNA-seq data for bantam-overexpressing cell clones mosaically induced in the eye-antennal discs.** (XLSX)

**S3 Table. Summary of statistical analyses.**
(XLSX)

**S1 Text. Detailed genotypes used in each figure.**
(DOCX)

**S2 Text. Supplementary methods.**
(DOCX)

## Acknowledgments

We thank M. Enomoto, Y. Sanaki, K. Taniguchi, and H. Kanda for discussions, Minoru Koijima, Miho Tanaka for technical assistance, and NGS core facility of the Graduate Schools of Biostudies, Kyoto University for supporting the RNA-seq analysis. We also thank M. Miura, T. Adachi-Yamada, D. Bilder, T. Xu, N. Perrimon, the Bloomington Stock Center, the National Institute of Genetics Stock Center (NIG-FLY), the *Drosophila* Genetic Resource Center (DGRC, Kyoto Institute of Technology), and the VDRC Stock Center for fly stocks, and Y. Grosjean for anti-JhI-21 antibody.

## Author Contributions

**Conceptualization:** Bojie Cong, Shizue Ohsawa, Tatsushi Igaki.

**Data curation:** Bojie Cong, Mai Nakamura, Yukari Sando, Takefumi Kondo, Shizue Ohsawa.

**Formal analysis:** Bojie Cong, Takefumi Kondo.

**Funding acquisition:** Shizue Ohsawa, Tatsushi Igaki.

**Investigation:** Bojie Cong, Mai Nakamura, Yukari Sando, Shizue Ohsawa.

**Methodology:** Bojie Cong.

**Project administration:** Tatsushi Igaki.

**Resources:** Tatsushi Igaki.

**Supervision:** Tatsushi Igaki.

**Visualization:** Bojie Cong, Shizue Ohsawa, Tatsushi Igaki.

**Writing – original draft:** Bojie Cong, Shizue Ohsawa, Tatsushi Igaki.

**Writing – review & editing:** Bojie Cong, Takefumi Kondo, Shizue Ohsawa, Tatsushi Igaki.

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
