## [Decision Letter · Decision Letter 0]

9 Oct 2020

Dear Tatsushi,

Thank you very much for submitting your Research Article entitled 'JNK and Yorkie drive tumor malignancy by inducing L-amino acid transporter 1 in Drosophila.' to PLOS Genetics. Your manuscript was fully evaluated at the editorial level and by independent peer reviewers. The reviewers appreciated the attention to an important problem, but raised some substantial concerns about the current manuscript. Based on the reviews, we will not be able to accept this version of the manuscript, but we would be willing to review again a much-revised version. We cannot, of course, promise publication at that time.

If you decide to revise the manuscript for further consideration at PLOS Genetics, please aim to resubmit within the next 60 days, unless it will take extra time to address the concerns of the reviewers, in which case we would appreciate an expected resubmission date by email to plosgenetics@plos.org.

[LINK]

We are sorry that we cannot be more positive about your manuscript at this stage. Please do not hesitate to contact us if you have any concerns or questions.

Yours sincerely,

Norbert Perrimon

Associate Editor

PLOS Genetics

Peter McKinnon

Section Editor: Cancer Genetics

PLOS Genetics

Reviewer's Responses to Questions

**Comments to the Authors:**

Reviewer #1: In this study, to characterize the common oncogenesis pathways, Cong et al compared differentially-expressed genes between RasV12/scrib-/- and bantam/rab5-/- larval disc tumors and identified JhI-21, a fly homolog of LAT1 that senses Leucine availability, as a crucial regulator for the growth of both of them. They further found that JhI-21 expression is commonly increased by JNK and yki signaling pathways to increase mTOR activity and promote growth in these two tumors. However, when authors fed larvae LAT1 inhibitors to try to inhibit Jhl-21 activity, they interestingly uncovered that the growth of only RasV12/scrib-/- tumors, but not bantam/rab5-/- tumors, was suppressed. Finally, combining RNA-seq and RNAi, they indicated the molecular mechanism(s) of drug resistance in bantam/rab5-/- tumors, whereby bantam suppresses the expression of CG31157, a TMEM135-like gene, to diminish effects of LAT1 inhibitors in the larval disc tumors.

Given the known fact that LAT1/2 mediates Leucine uptake, mTOR activation and tumor progression in mammals, this work provides novel findings of differential drug effects in two different tumors, mimicking genetic complexity and heterogenous nature of mammalian tumors. The authors also preliminarily indicated that bantam/CG31157 axis could be the key regulator of drug resistance, which could be possibly translated into future human cancer treatment. However, the current results of Leucine sensing, Jhi-21 inhibition, and CG31157 functions still are quite obscure. I recommend a major revision prior to the publication in Plos Genetics.

Major comments:

1. Multiple RNAi lines or mutants of JhI-21 and CG31157 should be used to exclude the off-target effects. In this study, only a single RNAi line for either JhI-21 or CG31157 was not sufficient to conclude the loss-of-function effects.

2. As indicated in Grosjean’s work, a poor medium supplied with Leucine is recommended to validate Leucine/JhI-21 axis in tumor growth regulation.

3. Although BCH has been previously shown to slow down Leucine uptake, the effects of Kyt-0353/JPH203 were not well evaluated in Drosophila. Because the drug concentrations in the fly food are so high (0.3 mg/mL BCH equals 2 mM, 0.2 mg/mL KYT-0353 equals 4 mM), different dosages should be tested.

On the other hand, the larvae were fed inhibitors simultaneously with tumor induction. The authors actually compared pS6 levels in the normal cells and well-expanded tumors, but not tumor responses to inhibitors. Therefore, the short-term feeding of drugs should be performed. For example, larvae are normally cultured for 6-9 days to develop disc tumors and then are fed drugs for 12 or 24 hours to evaluate pS6 levels in the tumors.

In vitro assays that treat isolated disc tumors with Leucine and different drugs are also recommended.

4. The drug effects in bantam/rab5-/- tumors with CG31157 overexpression should be examined.

Minor comments:

5. The mechanism(s) whereby JhI-21 expression is induced by JNK plus yki is quite interesting. However, the inducers for JNK (Eiger) and yki (yki.S168A) signaling pathways are weak ones and reflect neither signaling activations and progression in two tumors. Please used HepAC and yki.S111A.S168A.S250A. Otherwise, they can not conclude “JNK or yki activation alone did not induce JhI-21 expression”.

6. The relationships between CD98C/CG2791, Mnd, and JhI-21 have been studied to elucidate the functional regulation by heterodimeric complexes in leucine exchange in Drosophila. The authors should include them in both introduction and discussion sections to offer a general understanding of LAT physiology.

Reviewer #2: Igaki and colleagues use the power of Drosophila genetics and two different neoplastic tumor models to identify a role of JhI-21, a fly homolog of LAT1, a Leucine transporter, in tumor growth and malignancy. Authors use two different model systems of epithelial malignant transformation (first, the combination of an activated form of Ras: RasV12 and loss or apical-basal polarity determinants and, second, overexpression of a dominant negative form of Rab5 and overexpression of the bantam miRNA) to search, through transcriptional genomics, for genes upregulated in the two different conditions and whose depletion affetcs tumor growth. Authors make use of genetics and chemical inhibitors to unravel a role of LAT1 in tumor growth by promoting TOR signaling. Interestingly, authors identify a role of bantam miRNA in targeting CG31157, which encodes a TMEM135-like protein, and inducing resistance of these tumors to LAT1 inhibitors. The ms is well written, figures self-explanatory and the finding of LAT1 as a gene upregulated in fly tumors with a role in tumor growth will certainly contribute to the understanding of the role of the upregulation of this gene in human cancer. The identification CG31157 will also help identifying vulnerabilities of these tumors too. I have, though, two major concerns on the paper that should be addressed by the authors:

Major points:

(1) Antibody against JhI-21 should be validated in overexpression experiments (in tissues overexpressing Jlh-21 with the GAL4/UAS system). It was not validated in the original paper either. Authors should show by other means the upregulation of Jhl-21 in malignant cells (eg. in situ hybridization or GFP reporters?)

(2) JhI-21 knockdown affects the growth of the clones. I thus wonder whether the rescue of the growth of malignant clones could be a simple consequence of Jlh1 being required for cell survival or growth of normal cells. Have authors tried to use other JhI-21 RNAis not affecting the growth of control clones?

Minor points:

(3) Activation of the TOR pathway should be validated in western blots

(4) Code numbers in fly stocks should be included.

(5) A section on statistics should be added to M&M.

(6) A wild type eye should be added to Figure 1 for comparison.

Reviewer #3: This manuscript explores the control of cell growth in Drosophila neoplastic tumours. A strong upregulation of the TOR pathway is identified by pS6 staining, and this is attributed to a very mild upregulation of LAT1 amino-acid transporter.

The basic problem is that the magnitude of the induction of LAT1 is very, very small compared with the magnitude of the pS6 induction, which means that the authors have not identified the actual mechanism primarily responsible for inducing TOR activation in these tumours. Thus, I cannot support publication.

Reviewer #4: Cong and colleagues investigate tissue growth control in Drosophila imaginal discs. They use several well studied genetic combinations (RAS, Hippo, scribble, Rab5, bantam) that perturb cell polarity and proliferation. The power of the study is the unbiased RNA-sequencing experiment and subsequent mini genetic screen where they identify genes that can partially reverse the epithelia tumor overgrowth that is caused by perturbing these factors. They identify the Juvenile hormone Inducible-21 (JhI-21), a Drosophila homolog of the L-amino acid transporter 1 (LAT1), as a gene that is elevated in the tumors and required for their growth. They propose that JNK and the Hippo protein Yorkie drive its expression and that Jhl-21 induces the Tor pathway to drive tumor growth. They further explore the role of the microRNA bantam in these tumors.

The study is interesting and the screen is powerful. They use many approaches and experiments that they and others have employed in other studies, and often repeat published results (e.g. warts expression can suppress RAS/scribble tumors). They have some interesting observations that specialists within the Drosophila tumor field will appreciate. At this stage I don’t think the manuscript will be of substantial interest to non-specialists. The major shortfall of the study is that the mechanism they have out forward for how the different factors intersect (Hippo, polarity, Jhl-21, Tor etc. is not well substantiated and hard to reconcile. The paper reads more like a conglomeration of interesting observations than a pithy dissection/discovery of a new feature of epithelial tissue growth. However, I think the paper will help to shape the studies of others in the fly epithelial tissue growth field, although the manuscript requires far more effort to properly present and discuss the data. In particular, the authors to validate their Jhl-21 experiments by using independent RNAi line and/or mutants, and provide validation of the Jhl-21 antibody. Comments that should help to improve the manuscript are:

Introduction:

This needs more text to properly introduce the study. Also, it jumps too abruptly from para 1 to para 2. Given that LAT1 was found in an unbiased fashion it is probably better introduced at the end of the introduction.

Similarly, the Discussion is very brief and does not adequately discuss the findings of the study, their significance and limitations and important areas to address in the future.

The Jhl-21 RNAi line used should be verified with protein expression and/or RNA experiments, to prove that it targets Jhl-21 expression. They can verify the Jhl-21 antibody in imaginal discs and the RNAi line in one experiment.

At least one other Jhl-21 RNAi line should be used to verify the experiments presented in this paper. Alternatively, a mutant allele should be used.

Rheb mutant alleles impair cell growth and proliferation. Why does Rheb RNAi give no phenotype in Figure 3. Has it been proven to deplete Rheb levels? If not, this should be done.

**Have all data underlying the figures and results presented in the manuscript been provided?**

Reviewer #1: Yes

Reviewer #2: Yes

Reviewer #3: Yes

Reviewer #4: Yes

PLOS authors have the option to publish the peer review history of their article (what does this mean?). If published, this will include your full peer review and any attached files.

Reviewer #1: No

Reviewer #2: No

Reviewer #3: No

Reviewer #4: No

---

## [Decision Letter · Decision Letter 1]

22 Jun 2021

Dear Dr Igaki,

Thank you very much for submitting your Research Article entitled 'JNK and Yorkie drive tumor malignancy by inducing L-amino acid transporter 1 in Drosophila.' to PLOS Genetics.

The manuscript was fully evaluated at the editorial level and by independent peer reviewers. The reviewers appreciated the attention to an important problem, but raised some substantial concerns about the revised manuscript. Based on the reviews, we will not be able to accept this version of the manuscript, but we would be willing to review a much-revised version. We cannot, of course, promise publication at that time. The three significant issues that need to be addressed are: 1.  Statistics; 2. Specificity of the antibody against JhI-21; and 3. the strongly increased TOR activation attributed to the  very mild upregulation of LAT1 in tumor. We encourage you to provide a revised manuscript that addresses these important issues.

If you decide to revise the manuscript for further consideration at PLOS Genetics, please aim to resubmit within the next 60 days, unless it will take extra time to address the concerns of the reviewers, in which case we would appreciate an expected resubmission date by email to plosgenetics@plos.org.

[LINK]

We are sorry that we cannot be more positive about your manuscript at this stage. Please do not hesitate to contact us if you have any concerns or questions.

Yours sincerely,

Norbert Perrimon

Associate Editor

PLOS Genetics

Peter McKinnon

Section Editor: Cancer Genetics

PLOS Genetics

Reviewer's Responses to Questions

**Comments to the Authors:**

Reviewer #1: I think the authors have addressed all of my comments and recommend the acceptance for this manuscript.

Reviewer #2: Authors have addressed satisfactorily some of my previous concerns. I still have two major issues:

(!) I do believe authors should take statistics more seriously. First, Student t-test statistical analysis is not correct in most cases and authors should use alternative ones such as Dunnet's due to the presence of more than one experimental set. Second, authors should include a supplementary table where all data are summarised: n, p values, statistical analysis used, mean, standard deviation and p values for all figure panels where a quantification is being done. Third, authors should explicitly explain which program was used to perform the statistical analysis and say whether all experiments were done the same time. Fourth, I wonder whether the same Ras/scrib control was used to analyse the impact of two RNAis to rescue clone size in Figure 1o. I hope the answer is no. If so, two different control bars should be included or the second set of experiments where RAS/scrib clone size was compared to the new set of clones in the presence of the new RNAi.

(2) I am not convinced with the answer to my concern on the antibody against JhI-21. Have authors tested whether the protein can be detected with an antibody to HA: when the UAS-JhI-21.ORF.3xHA.GW transgene is over expressed with the GAL4/UAS system? There are EP-lines available. Can authors test in this case?

Reviewer #3: The authors still didn't address the basic problem. It is no surprise that you need amino acid transport for TOR activation. The problem is that they attribute the strongly increased TOR activation in their tumours to a very mild upregulation of LAT1, which is just not a well supported conclusion.

**Have all data underlying the figures and results presented in the manuscript been provided?**

Reviewer #1: Yes

Reviewer #2: Yes

Reviewer #3: Yes

PLOS authors have the option to publish the peer review history of their article (what does this mean?). If published, this will include your full peer review and any attached files.

Reviewer #1: No

Reviewer #2: No

Reviewer #3: No

---

## [Decision Letter · Decision Letter 2]

19 Oct 2021

Dear Dr Igaki,

We are pleased to inform you that your manuscript entitled "JNK and Yorkie drive tumor malignancy by inducing L-amino acid transporter 1 in Drosophila." has been editorially accepted for publication in PLOS Genetics. Congratulations!

Yours sincerely,

Norbert Perrimon

Associate Editor

PLOS Genetics

Peter McKinnon

Section Editor: Cancer Genetics

PLOS Genetics

Comments from the reviewers (if applicable):

Reviewer's Responses to Questions

**Comments to the Authors:**

Reviewer #2: I thank authors for having taken the statistics more seriously after this round of revision. I congratulate them for this nice paper.

Reviewer #3: The competing manuscript pointed out by the authors makes a strong case the LAT1 upregulation in APC mutant tumours is a conserved phenomenon. This strengthens the importance of their observations in Drosophila.

**Have all data underlying the figures and results presented in the manuscript been provided?**

Reviewer #2: Yes

Reviewer #3: Yes

PLOS authors have the option to publish the peer review history of their article (what does this mean?). If published, this will include your full peer review and any attached files.

Reviewer #2: No

Reviewer #3: No

**Data Deposition**

http://datadryad.org/submit?journalID=pgenetics&manu=PGENETICS-D-20-01337R2

**Press Queries**

---

## [Editor Report · Acceptance letter]

9 Nov 2021

PGENETICS-D-20-01337R2 

JNK and Yorkie drive tumor malignancy by inducing L-amino acid transporter 1 in Drosophila 

Dear Dr Igaki, 

We are pleased to inform you that your manuscript entitled "JNK and Yorkie drive tumor malignancy by inducing L-amino acid transporter 1 in Drosophila" has been formally accepted for publication in PLOS Genetics! Your manuscript is now with our production department and you will be notified of the publication date in due course.

With kind regards,

Anita Estes

PLOS Genetics

On behalf of:
